# Orofacial Manifestation of Systemic Sclerosis: A Cross-Sectional Study and Future Prospects of Oral Capillaroscopy

**DOI:** 10.3390/diagnostics14040437

**Published:** 2024-02-16

**Authors:** Anna Antonacci, Emanuela Praino, Antonia Abbinante, Gianfranco Favia, Cinzia Rotondo, Nicola Bartolomeo, Massimo Giotta, Florenzo Iannone, Germano Orrù, Maria Teresa Agneta, Saverio Capodiferro, Giuseppe Barile, Massimo Corsalini

**Affiliations:** 1Complex Operative Unit of Odontostomatology, Department of Interdisciplinary Medicine, University of Bari “Aldo Moro”, 70124 Bari, Italy; anna_antonacci@yahoo.it (A.A.); antonella.abbinante@gmail.com (A.A.); gianfranco.favia@uniba.it (G.F.); agneta.mt.mta@gmail.com (M.T.A.); saverio.capodiferro@uniba.it (S.C.); massimo.corsalini@uniba.it (M.C.); 2Rheumatology Unit, Department of Precision and Regenerative Medicine and Ionian Area (DiMePReJ), University of Bari “Aldo Moro”, 70124 Bari, Italy; florenzo.iannone@uniba.it; 3Rheumatology Unit, Department of Medical and Surgical Sciences, University of Foggia, 71122 Foggia, Italy; cinzia.rotondo@gmail.com; 4School of Medical Statistics and Biometry, Department of Interdisciplinary Medicine, University of Bari “Aldo Moro”, 70124 Bari, Italy; nicola.bartolomeo@uniba.it (N.B.); massimo.giotta@uniba.it (M.G.); 5Department of Surgical Science, University of Cagliari, 09124 Cagliari, Italy; orru@unica.it

**Keywords:** scleroderma, capillaroscopy, oral health, quality of life

## Abstract

Background and objectives: oral alterations in Systemic Sclerosis (SSc) patients are widespread and include microstomia, periodontitis, telangiectasias, mandibular resorption, bone lesions, and xerostomia. This cross-sectional study aims to evaluate the differences between SSc patients (cases) and healthy subjects (controls) regarding oral manifestations, quality of life (QoL), and microcirculation alterations. Methods: plaque index (PCR), periodontal index (PSR), DMFT, salivary flow rate, and buccal opening were measured by expert clinicians. S-HAQ test, the Self-Rating Anxiety State (SAS), the Self-Rating Depression Scale (SDS), and the WHOQOL-BREF test were administered to patients to evaluate their QoL. Microvascular alterations were assessed by oral videocapillaroscopy, performed on gingival and labial mucosa. A statistical analysis was conducted to find significant differences between healthy people and SSc patients. Results: 59 patients were enrolled in this study. Standard salivary flow is significantly more frequent in controls, while xerostomia, reduced flow, microstomia, lip retraction, and periodontitis are significantly more frequent in the cases. Gingival capillaroscopy showed differences concerning loop visibility, thickening of the gum, tortuosity of gingival loops, and reduced gingival density. Labial capillaroscopy demonstrates that visibility of the labial loops, the labial ectasias, and the tortuosity of the loops are significantly associated with the presence of scleroderma. Hand and facial deformities, hypomobility of the tongue, cheeks, lips, microstomia, and xerostomia significantly compromised the quality of life of SSc patients, which was significantly worse among them. Moreover, oral videocapillaroscopy could be a proper diagnostic method to detect oral microcirculation alterations. SSc patients often present ectasias, rarefaction of the reticulum, microhemorrhages, and megacapillaries, which negatively impact their oral health. Conclusions: periodontitis, reduced salivary flow, and microstomia could be considered SSc oral manifestations. Joint deformities, facial appearance, and comorbidities significantly reduce the QoL of SSc patients compared to healthy subjects. Oral videocapillaroscopy could be an innovative and reliable technique to detect oral microcirculation anomalies.

## 1. Introduction

Systemic Sclerosis (SSc) is a chronic, multisystemic, and rare autoimmune disease that manifests itself with progressive sclerosis of tissues, from small fibrotic plaques to multiple inner organ involvement [1]. It affects mostly women within the fifth decades of their lives and has a variable incidence due to geographical clustering [2]. SSc is characterized by obliterative vasculopathy of small arteries, dysregulated immune system activation, and tissue remodeling, leading to skin and internal organ fibrosis, especially in the esophagus, lungs, and kidneys [3]. The etiology is not entirely known, and it is still under debate. It could be due to genetic predisposition and trigger factors such as chronic trauma, surgery, viral infections, and pharmaceutics [4]. Indeed, the increase in oxidative stress plays a pivotal role in the pathogenesis of SSc because Reactive Oxidizing Species (ROS) levels were higher in SSc patients [5]. Superoxide anion, hydroxyl radicals, nitric oxide, peroxynitrite, and hypochlorous acid are associated with the expression of pro-inflammatory and pro-fibrotic cytokines [6]. Therefore, the oxidative stress results in the activation of keratinocytes and T lymphocytes leads to the expression of fibroblast growth factor (FGF), tumor necrosis factor-α (TNF-α), transforming growth factor β (TGF-β), and platelet-derived growth factor (PDGF) with consequent aberrant production of extracellular matrix and increased collagen deposition [7]. In SSc, the progressive and multifocal visceral involvement may result in a reduced quality of life and life expectancy that clinicians cannot underestimate. Common SSc symptoms such as pain, gastrointestinal problems, joint deformities, movement limitation, dyspnea, dysphagia, and sleep disturbances are often related to psychiatric disorders such as anxiety and depression because of the chronic and progressive nature of SSc with a consequent dissatisfaction with their own appearance and for general disability condition [8]. Moreover, SSc induces aesthetically functional modifications of the hands, mouth, and face (microstomia, dental alteration, tongue hypomobility, and xerostomia) that can generate frustration about the individual body image. Oral alterations in SSc patients are widespread and include microstomia, enlargement of the periodontal ligament, telangiectasias, mandibular resorption, bone lesions, and xerostomia [9]. Actually, the diagnostic criteria of SSc were defined in 2013 by ACR/EULAR and include the presence of limited (lcSSc) or diffuse (dcSSc) skin thickening, development of digital ulcers and pitting scars, specific autoantibodies against centromere, topoisomerase-1 and RNA polymerase III, and the presence of “scleroderma spectrum disorders” at the nailfold capillaroscopy [10]. SSc patients may present with giant capillaries, microhemorrhages, avascularity, and neoangiogenesis at the nailfold videocapillaroscopy [11], which is a non-invasive, simple, fast, and reliable technique to evaluate the microcirculation; it is useful in differentiating SSc alteration from the primary Raynaud phenomenon [12,13]. Capillaroscopy could also be performed on oral mucosa in order to evaluate the local microcirculation abnormalities during diabetes, Sjogren syndrome, Behçet disease, Systemic Lupus Erythematous, Myositis, and SSc, but few studies have been conducted on the latter [14]. Therefore, the principal aim of this cross-sectional study is to evaluate the presence of oral manifestations and microcirculation alterations in SSc patients, their level of manual functional skills, and the influence of the disease on their quality of life. The secondary objectives are to establish an individual preventive/therapeutic protocol for relieving symptoms and delaying the progression of oral pathologies and to assess the perspective of oral capillaroscopy for monitoring oral health.

## 2. Materials and Methods

The study was conducted on patients referred to the Complex Unit of Odontostomatology, Interdisciplinary Department of Medicine, and Rheumatology Unit, Department of Precision and Regenerative Medicine and Ionian Area (DiMePRe-J), University of Bari. To be included in this study, the patients had to be over 18 years old, sign the informed consent, and participate actively in every follow-up visit. The exclusion criteria for both groups were the presence of a diagnosis of other autoimmune diseases, pregnancy, the presence of neoformation of the oral cavity, patients who did not sign the informed consent, and patients who did not complete every scheduled follow-up visit. The study was conducted in accordance with the Declaration of Helsinki, and their protocol was fully approved by the Local Ethical Committee of Hospital “IRCCS Giovanni Paolo II” (Study n. 1355/CE).

A cross-sectional study was set, and patients were divided into the following groups:Group A (cases) included all patients diagnosed with SSc;Group B (controls) consisted of all subjects without SSc diagnosis.

Group A included the patients with SSc diagnosis. SSc was classified and diagnosed according to the 2013 ACR/EULAR classification criteria for systemic sclerosis [10].

Group B included the subjects in healthy conditions, without other general comorbidities, and without daily drug therapy, in order to avoid the bias of other disease’s oral manifestations.

Procedures:

Data regarding demographic and clinical conditions were collected for each patient. During the first visit, the following aspects were investigated: family history of autoimmune diseases (year of diagnosis and age of onset of symptoms), oral hygiene habits, lifestyle, and presence of any other oral manifestations. A questionnaire was used to gather information on the presence of gingival bleeding, gingivitis, periodontal disease, dental mobility, reduced salivary secretion, halitosis, limited mouth opening, lip retraction, and oral breathing.

To perform an objective evaluation of orofacial features, the clinicians conducted these assessments as follows:PCR plaque index (Plaque Control Record) to evaluate the presence of plaque deposits on the tooth surfaces.Periodontal screening PSR (Periodontal Screening and Recording) to record the maximum probing depth for each sextant.DMFT (Decayed Missing and Filled Permanent Teeth) score to calculate the number of decayed, missing, and filled teeth.Salivary flow quantification was conducted by collecting a saliva sample and determining the buffering power (pH) using reactive strips with a colorimetric change.The degree of microstomia was measured by determining the distance between the incisal margins of the lower central tooth and the upper central tooth during the maximum oral opening.

The authors distinguished 3 different degrees of microstomia: mild (between 40 mm and 36 mm), moderate (between 35 mm and 30 mm), and severe (less than 29 mm). Moreover, the other part of this study consists of multiple questionnaires regarding the ability of the upper and lower limbs, the anxiety and depression level, and patients’ quality of life. Functional abilities of the upper and lower limbs were assessed using the s-HAQ test, which consists of 20 questions in different functioning categories, measured on a scale from zero (no difficulty) to three (severe difficulty). The patients were also subjected to the Self-Rating Anxiety State (SAS) and the Self-Rating Depression Scale (SDS) tests, each consisting of 20 questions with scores ranging from 1 to 4 to assess anxiety and depression. The patient’s quality of life was analyzed using the WHOQOL-BREF test, comprising 26 questions divided into four areas: physical health, psychological health, social relationships, and the environment. Each answer has a score from 0 to 5, and overall values are obtained from the average scores for each area.

Finally, oral capillaroscopy was used for local microcirculation evaluation as a non-invasive instrumental investigation that allows observation and photography of small vessels in vivo using a video microscope. The capillaroscopic investigation was conducted in the morning, consistently using the same light source, with patients in a sitting position. The areas investigated for each patient were the marginal gingiva and the labial mucosa corresponding to the lower incisors.

The conditions of the microcirculation were evaluated through parametric and non-parametric data:

Non-parametric data:Visibility of the loops: difficulty in focusing (score from 0 to 4).Orientation related to surface (score A, B, or AB). (A) pattern parallel to the surface; (B) pattern perpendicular; (AB) pattern both parallel and perpendicular.Microhaemorrhages: (Score 0 or 1). (0) absence; (1) presence.Characteristics of capillary loops (e.g., tissue thickening, ectasias, megacapillaries, and reticulum rarefaction): (score 0 or 1). (0) absence; (1) presence.

Parametric data:
Capillary density: the number of loops per surface unit (in a square millimeter, there are typically 12 to 16 loops present);Capillary tortuosity, measured on a scale from 0 to 1.

Follow-up

After the clinical data recording, each subject belonging to both groups was educated on how to improve their oral hygiene. Customized instructions were given to them to improve the manual toothbrush, the oral hygiene motivation, and the use of interdental floss or chlorhexidine mouthwash, especially in unpaired dexterity as SSc patients. General oral health was re-evaluated in follow-up visits performed at 1, 6, and 12 months. This follow-up mostly has a clinical implication and is unhelpful to scientific purposes because clinical data of periodic visits were not collected.

Statistical analysis

Continuous variables were expressed as mean and standard deviation (SD) for normally distributed parameters or the median and interquartile range (IQR) in the case of skewed data distribution. Shapiro–Wilk’s statistic was used to test normality. The distribution of patients in each category was described as frequency and proportion. Categorical variables are shown as frequency and percentage. Differences in continuous variables between cases and controls were compared using a Student’s *t*-test for normally distributed parameters or non-parametric Mann–Whitney U Test otherwise. Comparison for categorical variables was performed using the Chi-square test or Fisher’s exact test, as necessary. The analysis was conducted for homogeneous groups of variables: lifestyle, oral hygiene, oral cavity manifestations, and gingival and labial capillaroscopy parameters. Therefore, the parameters that were statistically significant in the univariate analysis were included in multivariable logistic regression models to test their independent effect on the probability of having scleroderma. Using the Akaike Information Criterion (AIC), a stepwise selection was used to estimate the final models. To reduce the bias of the estimates due to the low sample size, Firth’s penalized maximum likelihood estimation was used [15]. The logistic regression results are shown as Odds Ratio (OR) and their 95% confidence intervals. All tests of statistical significance were two-tailed, and *p*-values less than 0.05 were considered statistically significant. Statistical analysis was performed using the SAS/STAT^®^ Statistics, Version 9.4 (2013), SAS Institute Inc., Cary, NC, USA.

## 3. Results

The study includes 59 patients. They were divided into test group A, patients with a diagnosis of SSc (*n* = 25), and control group B, subjects without a diagnosis of SSc (*n* = 34). The two groups had no significant differences regarding smoking (*p* = 0.702) and sports (*p* = 0.262). Age was significantly higher in subjects with SSc (*p* = 0.021); among these, the consumption of snacks (*p* = 0.034) and the use of the manual toothbrush (*p* = 0.034) were significantly more frequent, while the use of the electric toothbrush (*p* = 0.034) was more frequent among the controls. The use of mouthwash was similar between SSc and controls (*p* = 0.943). With the multivariate logistic model, only the significance of age remains, with an Odds Ratio (OR) of 1.05 [CI95% 1.01–1.10]. An increase of one year of age corresponds to a 5% increase in the risk of being scleroderma. These results are reported as follows (Table 1).

Regarding the SSc orofacial manifestations, there was no statistically significant difference between cases and controls regarding halitosis (*p* = 0.057) and gingivitis (*p* = 0.153). Standard Flow was significantly more frequent in controls (*p* < 0.001), while xerostomia (*p* < 0.001), reduced flow (*p* = 0.004), microstomia (*p* < 0.001), lip retraction (*p* < 0.001), and periodontitis (*p* = 0.002) were significantly more frequent in the cases. After a multivariate logistic model including all significant variables and age, microstomy, specifically of a “mild” degree compared to absence (*p* = 0.002), and lip retraction (*p* = 0.001) remain significant (Table 2).

The PCR index is significantly higher in SSc (*p* = 0.001) than in the DMFT (*p* < 0.001). The PSR score is significantly worse in SS patients than in the controls (*p* < 0.001). The multivariate logistic model analysis (also adding age) highlights the independent effect of the DMFT (*p* = 0.012): for each additional decayed, filled, or missing tooth, there is more than double the risk of belonging to the group of scleroderma subjects (OR 2.19) (Table 3).

All parameters indicative of function, such as s-HAQ test (*p* < 0.001) and quality of life, with anxiety degree (*p* < 0.001), depression degree (*p* < 0.001), physical health (*p* < 0.001), psychological state (*p* < 0.001), social relations (*p* < 0.001), and environmental conditions (*p* < 0.001), show a condition significantly worse among the cases. Among the above-cited parameters, the only one that remains significant with the multivariate logistic model was the s-HAQ test (*p* = 0.001). On the other hand, among the scores, only the one relating to the environmental conditions remains significant (*p* < 0.001) (Table 4).

Regarding gingival capillaroscopy parameters, there is no significant difference between cases and controls (*p* = 0.137) for microhemorrhages. In contrast, loop visibility (*p* < 0.001), thickening of the gum (*p* = 0.002), tortuosity of gingival loops (*p* < 0.001), and reduced gingival density (*p* = 0.001) are significantly worse in the cases than in the controls. The multivariate model (including age) shows an independent association between the presence of scleroderma, the thickening of the gingival tissue, and the reduced gingival density. Regarding labial capillaroscopy, with the univariate analysis, the visibility of the labial loops (*p* < 0.001), the labial ectasias (*p* = 0.033), and the tortuosity of the loops (*p* < 0.001) are significantly associated with the presence of scleroderma, and with the multivariate logistic model, the significance of both visibility (*p* = 0.048) and tortuosity (*p* = 0.025) remains significant; together with these two parameters, advanced age also plays a role, albeit not significant (*p* = 0.063). There are no differences between cases and controls regarding lip microhemorrhages (*p* = 0.278), thickening of lip mucosa (*p* = 0.067), labial megacapillaries (*p* = 0.067), and rarefaction of the labial reticulum (*p* = 0.137) (Table 5).

## 4. Discussion

This cross-sectional study aimed to evaluate the differences regarding orofacial features, quality of life, and labial and gingival capillaroscopy between the patients affected by SSc and healthy controls. Several differences have been significant, which are described below.

There were not any differences in smoking habits between SSc patients and controls, but it is mandatory to consider that even if the adverse effect of smoking is well established in pulmonary efficiency, it does not affect the progression of SSc disease, denying the major role of tobacco in the etiopathogenesis of SSc [16]. Moreover, we found that the consumption of snacks was significantly higher in SSc patients compared to healthy people, but the diet influence on the SSc patients remains still unclear: a diet rich in short-chain fermentable oligosaccharides, disaccharides, monosaccharides, and polyol is not associated with a worsening of SSc gastrointestinal symptoms [17].

Oral hygiene conditions in scleroderma patients are compromised, confirming Isola G. et al. hypothesis [18], probably as a result of worsening of manual dexterity, which affects SSc in their late stages, especially concerning hand mobility and daily activities [19]. In this study, the SSc patients preferred the manual toothbrush to the electric one, with a significant difference. This trend, as reported by Yuen et al., should be inverted, as the electric toothbrush is related to a reduction of plaque and gingival inflammation indexes in SS patients, suggesting the use of the child-sized brush head, which can overcome the microstomia and reduce hand mobility due to sclerodactyly [20]. For this reason, dental floss use is very low among SS patients (8%) because it requires better hand mobility and coordination. This result is consistent with other studies, suggesting that flossing devices are recommended to increase the dental floss practice in SS patients [20,21]. In addition to the electric toothbrush, the use of mouthwash (only used by half of the SS patients group) has to be suggested in SS patients because it is considered a protective factor that can reduce the oral bacterial load and gingival indexes [22].

As reported in the literature, SSc may present several orofacial manifestations: the most reported are the characteristic “mask-face” due to the reduced elasticity and atrophy of facial skin, telangiectasias, and microstomia [21]. This study analyzed the differences between the SSc patients and the control group concerning oral features.

The salivary flow was the first parameter that has been investigated. We adopted the unstimulated saliva buffering PH power to establish the flow rate because it is the most reliable and non-invasive method at this time [23]. The xerostomia was evaluated in 60% of the SSc patients, which is significantly higher than the healthy people. In addition to the xerostomia, the reduced salivary flow was present in 24% of SSc patients, in contrast to no control group participant that does not show any salivary alterations. This finding is consistent with many authors, confirming the xerostomia and reduced salivary flow as a very common oral manifestation of SSc. Said et al. reported a rate of 63.4% [24], Couderc et al. reported a rate of 40% [25], and Crincoli et al. reported a rate of 78.8% prevalence of xerostomia in SSc patients [26]. An interesting study by Parat et al. concluded that the salivary flow rate is correlated to the SSc activity and severity, suggesting their monitoring to evaluate the SSc course progression [27]. However, the reduced salivary flow of SSc patients was often related to a secondary Sjogren Syndrome, but this direct association is still under debate [21,28]. The histopathologic features of minor salivary glands during SS disease are referred both to acinar fibrosis (related to the upregulation of the collagen deposition in SSc) and to autoimmune acinar destruction as well as the Sjogren Syndrome [29]. Even if is more frequent, halitosis is not significantly higher in SSc patients despite it being a common symptom of xerostomia or reduced salivary flow. Microstomia is defined as a limitation of mouth opening with a measured inter-incisive distance of less than 40 mm. As the xerostomia, microstomia is a usual orofacial manifestation of SS with a percentage between 60% and 91.4%, resulting in a strong correlation with SSc [30]. In this study on 59 patients, we report an overall percentage of 100% of SSc patients that presented microstomia, respectively, with 12% of severe degree (less than 29 mm), 48% of moderate degree (between 30 and 35 mm), and 40% of mild degree (between 36 and 40 mm). Microstomia was also reported in more than half of controls (55.9%). Our results are consistent with previous studies [21,26,31,32], suggesting a significant correlation between microstomia and SSc. Lip retraction and oral breathing have not been investigated in the literature. In our study, both are significantly higher compared to SSc patients, probably due to progressive fibrosis that could manifest as lip retraction in the initial stage, and subsequently, labial incompetence that could lead to mouth breathing [33], worsening the sicca syndrome mentioned above. The xerostomia, microstomia, and oral breathing have a major role in the development of dysphagia in SSc patients. Dysphagia is defined as the difficulty of swallowing and could be attributed to several causes. In SSc patients, dysphagia is mostly due to an impairment of the oral phase of swallowing because of xerostomia, microstomia, and loss of dental elements, which prevent proper bolus formation [34]. Moreover, many studies confirm that in the hypofunction of salivary glands, the harmony of the oral cavity is compromised [35,36]. An increase in microflora is observed, especially regarding streptococci, actinomycetes, and lactobacilli, which increases the likelihood of developing cavities, periodontal disease, cheilitis, and candidiasis [37,38]. In fact, this aspect was deepened in this study, first evaluating the presence of gingivitis and periodontitis and then recording the PCR, DMFT, and PSR scores. We found that periodontitis was significantly more frequent in SSc patients than gingivitis. As confirmed by Isola et al., despite the orofacial manifestations in SSc patients being well described in the literature, gingivitis and periodontitis indexes have been examined by a limited number of studies [18]. Our results are consistent with the authors who evaluated this aspect during SSc [18,39,40]. Different from other case–control studies that evaluate periodontitis by Probing Depth (PD), Attachment Loss (AL), and Bleeding on Probe (BOP), to the best of our knowledge, this is the first study that considered the PSR index to evaluate the periodontitis [41]. We decided to use this index due to its speed and ease of recording, which are essential in patients with limitations as SSc patients [42]. A possible explanation of such prevalence of periodontitis in SSc patients, besides the worse oral hygiene, could be in a reduced expression of TGF-β1 (an anti-inflammatory cytokine) in SSc patients, which is related to periodontal disease [43]. Moreover, the last index we evaluated was the DMFT, which has been evaluated in the other case–control studies. The mean DMFT score was significantly higher in the SSc patients than in the healthy group, with a *p* < 0.001, which indicates a strong correlation. This result is in contrast with the Zhang et al. meta-analysis, which reported no difference in DMFT score between SSc patients and controls [41]. Maybe we could explain this as a characteristic of SSc oral manifestation in the region where the study was conducted, located in the south of Italy, where the DMFT score among healthy people is usually very low [44]. After the data collection and diagnostic assessment, each patient was educated following a customized scheme of domiciliary oral hygiene procedures. The dental hygienists that participated in this study, after the periodontal screening and other oral disorder detection, in agreement with the dentist, formulated a personalized treatment plan to improve the already bad oral health of SSc patients, understanding their capabilities and their specific needs. Thus, according to other authors [20,45,46], tailored education management is mandatory, which takes into account the following interventions.

Our institution’s proposed management of the oral manifestations of SSc is reported in the table below (Table 6).

It is clear that the orofacial manifestations (e.g., the “mask face”, the reduction in salivary flow, the microstomia, and the progressive loss of teeth) negatively affect the overall quality of life. Many studies have evaluated this aspect in SSc patients, but only a few authors have focused on the impact of oral manifestation on quality of life in SSc patients [29,47]. In fact, they concluded that SS patients have significantly impaired oral health-related quality of life, suggesting the targeted interventions to improve their oral health and quality of life. As reported by Garaiman A. et al., 2021 [48], the psychological state of SSc patients is altered. The common symptoms, such as pain, gastrointestinal issues, joint deformities, limited mobility, dyspnea, and sleep disturbances, are associated with psychiatric disorders, such as anxiety and depression, as a direct consequence of the chronic nature of the disease. In accordance with other authors, we found that the s-HAQ questionnaire, anxiety degree, depression degree, physical health, psychological state, social relations, and environmental conditions are significantly worse in the SSc patients compared to controls.

The last part of this study evaluates the comparison between SSc patients and healthy people regarding gingival and labial capillaroscopy to assess if the latter could be considered a viable alternative to nailfold capillaroscopy in patients with systemic vascular diseases. Nailfold capillaroscopy is a safe and well-established method for evaluating structural alterations of the microcirculation. It is the gold standard in investigating and monitoring patients presenting Raynaud’s phenomenon [12,48,49,50,51]. The detection of the characteristic “scleroderma pattern” in nailfold capillaroscopy can be very helpful to SSc diagnosis and during the monitoring of the disease above all for its predictive value of digital ulcer risk [52,53,54]. Since it was described by Grassi et al. in 1993 [55], only Scardina et al. in 2005 [56] decided to evaluate the viability of oral capillaroscopy in different oral pathologies. Oral capillaroscopy is a non-invasive, harmless, repeatable examination with relatively low costs, enabling real-time monitoring of oral vascular lesions and the course of peripheral vasculopathy that aims to be a predictable diagnostic exam that evaluates the local microcirculation of oral mucosa. With the introduction of the technique of videocapillaroscopy with an optical polarized light probe, all the issues related to classical capillaroscopy methods have been overcome [57]. Digitalization has also ensured the standardization of parameters such as diameter, length, vessel morphology, capillary density, and blood flow, permitting their comparison over time. It is evident that this examination can also be performed in many research topics of dentistry within the mucosal surface of the oral cavity. Currently, oral capillaroscopy can be used for several diseases, such as Behçet Syndrome, Sjogren Syndrome, diabetic microangiopathy, periodontitis, COVID-19, and SSc [14,58,59,60,61]. The evaluation of vascular involvement in the oral cavity is crucial in the management of SSc patients. As confirmed by the study of Cutolo M. et al., 2019 [62], it is a progressive self-amplification process, inducing microvascular/endothelial damage, followed by autoimmune response, inflammation, and finally fibrosis. This aspect was supported by Sha et al., who reported a decreased oral microvasculature, demonstrated through a Sidestream Dark Field videocapillaroscopy on gingival tissues [63]. Consisting to the cited-above authors, we found that the SSc patients group showed a worse condition of oral microcirculation: loops visibility, thickening of the gum tissue, and gingival density indexes were significantly lower in SSc patients that also showed higher tortuosity of the gingival loops in gingival capillaroscopy. Similar results were evaluated in labial capillaroscopy, where loop visibility was significantly lower, and lip ectasia and tortuosity labial loops were significantly higher in SSc patients. After this result, we agree with other authors on the predictability of oral videocapillaroscopy as a diagnostic helpful tool to study peripheral microcirculation in healthy people and their pattern changes during systemic disease [59,60,63,64].

This study is not without limitations: the sample is a convenience sample, so the sample calculation was not conducted: it is a small sample that prevents drawing definitive conclusions. Another limitation is regarding the videocapillaroscopy, which is an operator-dependent diagnostic technique. In this case, the clinicians who performed videocapillaroscopy have decennial experience in videocapillaroscopy, so a correct training curve is mandatory. The future direction of this research is to further evaluate the viability of oral capillaroscopy to detect microcirculation alterations of oral mucosae and it will be interesting to compare the SSc oral manifestations with other autoimmune diseases, such as rheumatoid arthritis, other connective-tissue disorders, and autoimmune vasculitis that affect the oral cavity.

## 5. Conclusions

Hand deformities and oral hypomobility significantly compromise the domiciliary oral hygiene maneuvers of SSc patients, resulting in a bad oral condition. Critical screening indexes such as PCR, PSR, and DMFT are significantly worse in SSc patients compared to healthy people. Reduced salivary flow, microstomia, and lip retraction are more frequent in SSc patients, which, combined with progressive disability, constant appearance changes, and physical suffering, result in a decline in the overall quality of life. Anxiety and depression degrees are significantly higher in SSc patients, in contrast to physical and psychological state, social relations, and environmental conditions scores, which are lower compared to healthy subjects. Moreover, oral videocapillaroscopy could be suggested as a proper diagnostic non-invasive method to detect microcirculation alterations such as ectasias, rarefaction of the reticulum, microhemorrhages, and megacapillaries, which are more frequent in SSc patients and negatively impact their oral health. Dentists and dental hygienists play a crucial role in prevention and therapeutic interventions, collaborating closely with rheumatologists. It is mandatory to provide a tailored oral health management plan to SSc patient to improve their oral conditions.

## 6. Clinical Case 

A clinical case of a SSc patient with moderate microstomia and severe periodontitis was reported below (Figure 1 and Figure 2).

## Figures and Tables

**Figure 1 diagnostics-14-00437-f001:**
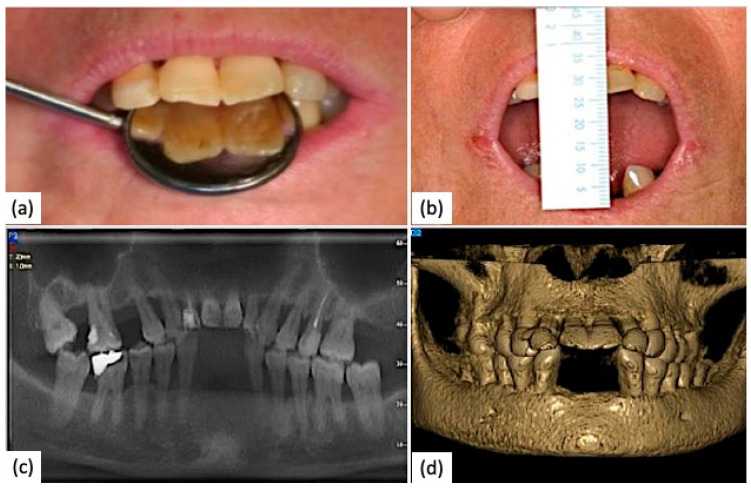
Oral manifestations in an SSc patient. (**a**,**b**) Moderate microstomia, with the presence of angular cheilitis (**b**); (**c**) orthopantomography showing diffuse periodontitis; (**d**) 3D TC cone beam showing mandibular and maxillary atrophy.

**Figure 2 diagnostics-14-00437-f002:**
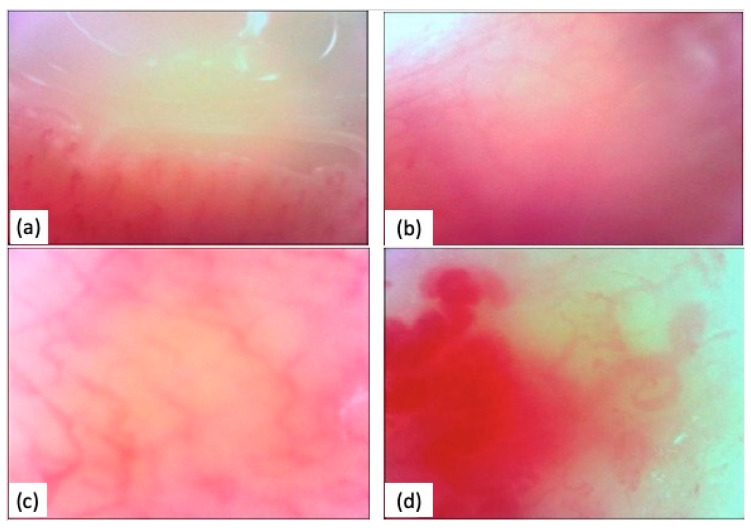
Oral videocapillaroscopy comparison. (**a**) Healthy gingival capillaroscopy; (**b**) sick gingival capillaroscopy; (**c**) healthy labial capillaroscopy; (**d**) sick labial capillaroscopy.

**Table 1 diagnostics-14-00437-t001:** Age, lifestyle, and oral hygiene. Comparison between cases and controls.

Parameters	Total (*n* = 59)	SSc (*n* = 25)	Controls (*n* = 34)	*p*
Age	50.3 ± 12.7	54.7 ± 12	47.1 ± 12.4	**0.021**
Smoke	12 (20.3)	4 (16)	8 (23.5)	0.702
Sport	15 (25.4)	4 (16)	11 (32.6)	0.262
Snacks	53 (89.8)	25 (100)	28 (82.4)	**0.034**
Manual toothbrush	45 (76.3)	23 (92)	22 (64.7)	**0.034**
Electric toothbrush	14 (23.7)	2 (8)	12 (35.3)	**0.034**
Dental floss	13 (22)	2 (8)	11 (32.4)	0.056
Tongue cleaner	4 (6.8)	1 (4)	3 (8.8)	0.603
Mouthwashes	31 (52.5)	13 (52)	18 (52.9)	0.943

**Table 2 diagnostics-14-00437-t002:** Manifestations of the oral cavity. Comparison between cases and controls.

Parameters	Total (*n* = 59)	SSc (*n* = 25)	Controls (*n* = 34)	*p*
**Salivary Flow**				
Xerostomia	15 (25.4)	15 (60)	0 (0)	**<0.001**
Reduced Flow	6 (10.2)	6 (24)	0 (0)	**0.004**
Standard Flow	27 (45.8)	4 (16)	23 (67.7)	**<0.001**
**Halitosis**	15 (25.4)	10 (40)	5 (14.7)	0.057
**Microstomia**				**<0.001**
Severe	3 (5.1)	3 (12)	0 (0)	
Moderate	31 (52.5)	12 (48)	19 (55.9)	
Mild	10 (17)	10 (40)	0 (0)	
Absent	15 (25.4)	0 (0)	15 (44.1)	
**Lip retraction**	20 (33.9)	20 (80)	0 (0)	**<0.001**
**Oral breathing**	17 (28.8)	15 (60)	2 (5.9)	**0.007**
**Gingival/periodontal diseases**				
Gingivitis	29 (49.2)	15 (60)	14 (41.2)	0.153
Periodontitis	22 (37.3)	15 (60)	7 (20.6)	**0.002**

Data were expressed as frequency (%).

**Table 3 diagnostics-14-00437-t003:** Clinical parameters. Comparison between cases and controls.

Parameters	Total (*n* = 59)	SSc (*n* = 25)	Controls (*n* = 34)	*p*
**PCR**	39.5 ± 19.5	52.7 ± 21.2	32.1 ± 14.2	**0.001**
**DMFT**	5 [4–7]	8 [6–11]	4.5 [3–6]	**<0.001**
**PSR Score**				**<0.001**
0	6 (11.5)	0 (0)	6 (17.7)	
1	12 (23.1)	0 (0)	12 (35.3)	
2	10 (19.2)	2 (11.1)	8 (23.5)	
3	16 (30.8)	9 (50)	7 (20.6)	
4	8 (15.4)	7 (38.9)	1 (2.9)	

**Table 4 diagnostics-14-00437-t004:** Functionality and quality of life. Comparison between cases and controls.

Parameters	Total (*n* = 59)	SSc (*n* = 25)	Controls (*n* = 34)	*p*
**s-HAQ test**				**<0.001**
Severe	5 (8.5)	5 (20)	0 (0)	
Moderate	6 (10.2)	6 (24)	0 (0)	
Mild	14 (23.7)	14 (56)	0 (0)	
Absent	34 (57.6)	0 (0)	34 (100)	
**Anxiety degree**				**<0.001**
Severe	4 (6.8)	4 (16)	0 (0)	
Moderate	11 (18.6)	11 (44)	0 (0)	
Mild	8 (13.6)	0 (0)	8 (23.5)	
Very Low	10 (17)	10 (40)	0 (0)	
Absent	26 (44)	0 (0)	26 (76,5)	
**Depression degree**				**<0.001**
Severe	4 (6.8)	4 (16)	0 (0)	
Moderate	14 (23.7)	14 (56)	0 (0)	
Mild	7 (11.9)	7 (28)	0 (0)	
Absent	34 (57.6)	0 (0)	34 (100)	
**Physical health**	67 [34–78]	33 [31–36]	76.5 [71–84]	**<0.001**
**Psychological state**	67 [42–77]	42 [39–46]	75.5 [68–80]	**<0.001**
**Social relations**	67 [33–78]	32 [31–34]	76.5 [71–80]	**<0.001**
**Environmental conditions**	70 [33–81]	32 [30–34]	79.5 [76–86]	**<0.001**

Data were expressed as frequency (%) or median [interquartile range].

**Table 5 diagnostics-14-00437-t005:** Gingival and labial capillaroscopy. Comparison between cases and controls.

Parameters	Total (*n* = 40)	SSc (*n* = 25)	Controls (*n* = 15)	*p*
**GINGIVAL CAPILLAROSCOPY**				
**Loops visibility**				**<0.001**
Minimal	11 (27.5)	11 (44)	(0)	
Medium	7 (17.5)	7 (28)	(0)	
Standard	22 (55)	7 (28)	15 (100)	
**Gingival microhemorrhages**	5 (12.5)	5 (20)	(0)	**0.137**
**Thickening of the gum tissue**	11 (27.5)	11 (44)	(0)	**0.002**
**Tortuosity of the gingival loops**	14 (35)	14 (56)	(0)	**<0.001**
**Reduced gingival density**	12 (30)	12 (48)	(0)	**0.001**
**LABIAL CAPILLAROSCOPY**				
**Loops visibility**				**<0.001**
Minimal	9 (22.5)	9 (36)	(0)	
Medium	9 (22.5)	9 (36)	(0)	
Standard	22 (55)	7 (28)	15 (100)	
**Lip microhemorrhages**	4 (10)	4 (16)	(0)	0.278
**Thickening of the lip tissue**	6 (15)	6 (24)	(0)	0.067
**Lip ectasia**	7 (17.5)	7 (28)	(0)	**0.033**
**Labial megacapillaries**	6 (15)	6 (24)	(0)	0.067
**Rarefaction of the labial reticulum**	5 (12.5)	5 (20)	(0)	0.137
**Tortuosity labial loops**	16 (40)	16 (64)	(0)	**<0.001**

**Table 6 diagnostics-14-00437-t006:** Management of SSc oral manifestations.

Periodontal/Carious Lesions Prevention	Hyposalivation/Xerostomia Treatment	Microstomia
Periodic checks, scheduled professional oral hygiene sessions	Topical and systemic salivary substitutes/stimulants, use of alcohol-free chlorhexidine 0.12% mouthwashes, polyenzymatic systems in the form of gels, mouthwashes, and toothpaste	Oral stretching
Reduction in the frequency of intake of cariogenic/acidic foods, meticulous oral hygiene	Frequent intake of water, use of chewing gum without added sugar with enzymes, xylitol	
Manual toothbrush with medium/soft filaments (ergonomic handle or with ball), electric toothbrush	Using petroleum jelly or lip balm to moisturize lips	

## Data Availability

Data are contained within the article.

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
