# Peer review of "Orofacial Manifestation of Systemic Sclerosis: A Cross-Sectional Study and Future Prospects of Oral Capillaroscopy"

_diagnostics, 2024, doi:10.3390/diagnostics14040437_

Round 1

Reviewer 1 Report

Comments and Suggestions for Authors

The article entitled "Orofacial manifestations of Systemic Sclerosis: a case-control study and future prospects of oral capillaroscopy" is very interesting and introduce a new diagnostic method to detect oral microcirculation alterations in Systemic Sclerosis. For these reasons it deserves to be published in "Diagnostics".

The text of manuscript should be justified.

Line 44. A period should be placed after the sentence.

The abnormal oxidative stress as a pathogenetic trait of SSc should be addressed in the introduction (line 50-55).

Author Response

Dear Reviewer 1,

Thank you very much for your appreciating words. You can find our point-by-point reply to your valuable comments. We hope you could appreciate our efforts.

Best regards

Reviewer 2 Report

Comments and Suggestions for Authors

The subject matter carries clinical relevance to clinicians. However, the presentation, methods and discussion require some revision.

1. Abstract is not appropriate - it reads like an intro to a review article. The abstract should contain objectives, methods, population and specific results.

2. Study design is not case-control: from what I gathered from the methods, the study is a cross-sectional study that compares SSc patient to non-SSc patients. It is not clear to me what was the point of follow-up?

3. Control group must be defined. Are they healthy control, or are they patients with other autoimmune disease. It is not clear who these people are, and why they came to the clinic in the first place.

4.How was SSc diagnosed?

5.comparison of oral manifestation of SSc to a non-SSc group seemed redundant because these manifestation are already known to be associated with SSc. It is like comparing joint manifestation between rheumatoid arthritis patient to a healthy control - of course the results will be positive. Therefore, I think it would make more sense to compare SSc to other autoimmune rheumatic diseases. 

6.Table 6 and discussion on how to treat oral manifestation of SSc is irrelevant to the study and were not supported by study results. These should be removed. If the author want to write about treatment of oral manifestation in SSc patients, it would be more appropriate to write a separate review article.

7.The logistic regression analysis was mentioned in the methods but were not reported.

8.conclusion should be rewritten to better reflect the study findings rather than generic statements on SSc and its therapy

Comments on the Quality of English Language

some language editing is recommended

Author Response

Dear Reviewer 2,

Thank you very much for your appreciating words. You can find our point-by-point reply to your valuable comments in the attached PDF. We hope you could appreciate our efforts to improve the quality of our work, following your helpful suggestions.

Best regards.

Reviewer 3 Report

Comments and Suggestions for Authors

Dear Authors,

The article entitled "OROFACIAL MANIFESTATION OF SYSTEMIC SCLEROSIS: A CASE-CONTROL STUDY AND FUTURE PROSPECTS OF ORAL CAPILLAROSCOPY" comprises an interesting case-control study with an exhaustive evaluation of oral cavity alterations.

The article needs minor revision and these are my recommendations:

Add in Material and Methods the period of the patients' referral- lines 86-87.  It is important to mention the time since the SSc diagnosis was established, the type limited/diffuse, and the current therapy. 

Also, detail the selection criteria of the control group. In line 178 the number of patients in the control group is different from Table 1.

What about the oral mucosa lesions and tongue alterations mentioned in the MM chapter? Can you detail the presence of oral mucosa candidosis or angular cheilitis (present also in the clinical picture)? At least the percentage. 

In the Discussion Chapter include a paragraph on the study limitations.

Best regards!

Author Response

Dear Reviewer 3,

Thank you very much for your appreciating words. You can find our point-by-point reply to your valuable comments in the attached PDF. We hope you could appreciate our efforts.

Best regards.

Round 2

Reviewer 2 Report

Comments and Suggestions for Authors

Thank you for the revisions. The manuscript has improved substantially compared to the original version.

Comments on the Quality of English Language

no specific comment